# Ultra-Early Screening of Cognitive Decline Due to Alzheimer’s Pathology

**DOI:** 10.3390/biomedicines11051423

**Published:** 2023-05-11

**Authors:** Pengxu Wei

**Affiliations:** 1Beijing Key Laboratory of Rehabilitation Technical Aids for Old-Age Disability, Key Laboratory of Neuro-Functional Information and Rehabilitation Engineering of the Ministry of Civil Affairs, National Research Center for Rehabilitation Technical Aids, Beijing 100176, China; pengxuwei@buaa.edu.cn; 2Key Laboratory of Biomechanics and Mechanobiology, Ministry of Education, Beijing Advanced Innovation Center for Biomedical Engineering, School of Biological Science and Medical Engineering, Beihang University, Beijing 100083, China

**Keywords:** Alzheimer’s disease, dementia, ultra-early screening, cognitive decline

## Abstract

Alzheimer’s pathology can be assessed and defined via Aβ and tau biomarkers. The preclinical period of Alzheimer’s disease is long and lasts several decades. Although effective therapies to block pathological processes of Alzheimer’s disease are still lacking, downward trends in the incidence and prevalence of dementia have occurred in developed countries. Accumulating findings support that education, cognitive training, physical exercise/activities, and a healthy lifestyle can protect cognitive function and promote healthy aging. Many studies focus on detecting mild cognitive impairment (MCI) and take a variety of interventions in this stage to protect cognitive function. However, when Alzheimer’s pathology advances to the stage of MCI, interventions may not be successful in blocking the development of the pathological process. MCI individuals reverting to normal cognitive function exhibited a high probability to progress to dementia. Therefore, it is necessary to take effective measures before the MCI stage. Compared with MCI, an earlier stage, transitional cognitive decline, may be a better time window in which effective interventions are adopted for at-risk individuals. Detecting this stage in large populations relies on rapid screening of cognitive function; given that many cognitive tests focus on MCI detection, new tools need to be developed.

## 1. Introduction

Alzheimer’s disease (AD) is the leading cause of dementia in the elderly [1,2]. Patients with dementia present with memory deficits and other cognitive impairments and require assistance with daily life activities [3]. Alzheimer’s disease is often associated with comorbidities that aggravate the course of the disease. For instance, patients with type 2 diabetes mellitus have an increased risk of developing Alzheimer’s disease, and glucose metabolism abnormalities are frequent among patients with Alzheimer’s disease [4]. Cerebral and heart vascular pathologies are pathogenic contributors to age-related dementia including Alzheimer’s disease, being inextricably linked to disease onset and progression [5]. Both lower (<55 years) and older (≥55 years) ages of people with hypertension, with longer or shorter duration of diagnosis, were associated with cognitive decline in different domains [6]. Systemic infection is an important contributor to Alzheimer’s disease and vascular dementia and thus requires early identification and treatment in the elderly population [7]. Consequently, the contribution of these types of conditions should be considered in preventive and therapeutic approaches to address the health challenges of Alzheimer’s disease.

The 2018 National Institute on Aging—Alzheimer’s Association (NIA-AA) research framework defined Alzheimer’s disease by its pathologic processes documented by biomarkers, i.e., aggregated amyloid β (Aβ), pathologic tau and neurodegeneration, rather than by clinical symptoms/signs of the disease; Alzheimer’s disease refers to positive Aβ plaques and pathologic tau deposits, whereas Alzheimer’s pathologic change, as the earlier stage, exhibits abnormal Aβ biomarkers but normal pathologic tau biomarkers [3].

The International Working Group recommends that Alzheimer’s disease is a clinical–biological entity defined by a specific clinical phenotype associated with in vivo Aβ and tau biomarkers; thus, Alzheimer’s disease should be diagnosed for individuals with both positive biomarkers and specific clinical manifestations of Alzheimer’s disease. Individuals who are biomarker-positive but cognitively unimpaired should be considered at risk of progression to Alzheimer’s disease. The recommendation is different from the purely biological definition of Alzheimer’s disease proposed by the 2018 NIA-AA research framework [8]. However, both the 2018 NIA-AA research framework [3] and the 2021 recommendations of the International Working Group [8] emphasize that the pathology of Alzheimer’s disease is defined by Aβ and tau biomarkers and the pathogenesis of Alzheimer’s disease is a continuous progression process ranging from cognitively unimpaired people to individuals with severe dementia.

## 2. The Preclinical Stage of Alzheimer’s Disease Is a Long Period

Accumulating evidence indicates that Alzheimer’s disease is a continuum in that cognitive impairment in Alzheimer’s disease occurs continuously over a long period and the aggregation of Aβ deposits and tau protein in the brain is also a continuous process beginning before symptom onset [3,9,10,11,12,13,14,15,16].

Mild cognitive impairment (MCI) is a stage preceding dementia and defined as cognitive deficits that do not affect the functions of daily living. However, cognitive impairment occurs before MCI in the progression process of Alzheimer’s disease; this stage is termed pre-MCI in some of the literature, defined as subjective cognitive complaints that may hardly be assessed with objective neuropsychological tests [17,18,19]. Usually, familial Alzheimer’s disease is more severe and presents an earlier onset when compared with sporadic Alzheimer’s disease [20,21]. Mutations in presenilin 1 (PSEN1) are the leading cause of familial Alzheimer’s disease [22]. To identify a complete progression course of Alzheimer’s disease, 449 individuals with a PSEN1 mutation were followed for 15 years. Cognitive impairment was detected approximately two decades before dementia onset in the PSEN1 carriers. The cognitive deficits were mainly amnestic but occurred also in other cognitive domains; transient recovery during symptomatic pre-MCI could be observed whereas continuous cognitive decline followed. Among the PSEN1 carriers, five stages in the progression course were classified: asymptomatic pre-MCI, symptomatic pre-MCI, MCI, dementia and death (Figure 1) [19].

The aggregation of Aβ deposits and tau protein is a continuous process. In the early stage of Aβ accumulation, Aβ deposits in the brain occur in the basal portions of the frontal, temporal, and occipital lobes. The presubiculum and the entorhinal layers show weakly stained Aβ whereas no Aβ is found in the hippocampus. Afterward, medium Aβ is present in almost all isocortical association areas except the primary sensorimotor cortices. Hippocampal formation is slightly involved. In the end stage, dense Aβ deposits appear nearly in all isocortical areas. Hippocampal formation exhibits relatively few Aβ deposits. A number of subcortical nuclei are also involved. The early stages of tau-positive neurofibrillary changes occur in transentorhinal areas; afterward, the transentorhinal region and entorhinal cortex are remarkably affected, and the end stages demonstrate severe tau pathology in the isocortex [23,24].

A transmembrane protein, the amyloid precursor protein (APP), is cleaved by α- and γ-secretases to generate nonpathogenic fragments. Aβ peptides are formed if APP undergoes sequential cleavage by β- and γ-secretases [25,26]. Aβ (1–40) and Aβ (1–42), also referred to as Aβ40 and Aβ42, are thought to be the most important Aβ isoforms. Compared with Aβ40, Aβ42 has two extra amino acids at the C-terminal region [27]. Aβ40 is more abundant whereas Aβ42 presents much higher toxicity and a greater aggregation tendency [28]. Compared with Aβ42, aggregates containing Aβ43 (a longer Aβ C-terminal variant) are more likely to induce cerebral Aβ deposition [29]. Longer Aβ variants (Aβ42 and Aβ43) tend to be detected in Alzheimer’s disease patients, whereas shorter forms (Aβ37, Aβ38 and Aβ40) are more likely to be found in subjects with intact cognition [30,31,32]. Apolipoprotein E (APOE) binds and transports Aβ across the blood–brain barrier into the blood, thereby facilitating perivascular Aβ clearance. Human APOE is polymorphic with the *ε*2, *ε*3 and *ε*4 alleles that encode the APOE isoforms APOE2, APOE3 and APOE4 [33]. Among the APOE isoforms, APOE4 has the least efficiency at binding and transporting Aβ, which causes gradual Aβ accumulation in the brain. The allele *ε*4 of APOE4 (APOE *ε*4) is the most prevalent genetic risk factor for sporadic Alzheimer’s disease [34]. APOE3 is neutral whereas APOE2 is protective [33].

Aβ depositions precede the development of tau pathology in some individuals whereas tau pathology may precede the deposition of Aβ in other cases [23,24]. Six tau isoforms are expressed in the adult human brain; three isoforms have three carboxy-terminal repeat domains (3R tau) and the other three isoforms have four repeats (4R tau). Physiologically, tau protein is highly soluble and shows little tendency for aggregation [35]. In Alzheimer’s disease, 3R + 4R tau aggregations (tauopathies) are found in the filaments and neurofibrillary tangles [35,36,37]. Both Aβ deposition and aggregation of hyperphosphorylated tau cause neurodegeneration and cognitive impairment [38].

Alzheimer’s disease can be classified into three variants according to genetic backgrounds, environmental factors, and the burden of Aβ/tau pathology, neurodegeneration, and cognitive symptoms: autosomal dominant Alzheimer’s disease, APOE ε4-related sporadic Alzheimer’s disease and APOE *ε*4-unrelated sporadic Alzheimer’s disease [34]. Marked cognitive impairments often manifest at approximately 50 years old in autosomal dominant Alzheimer’s disease, 75 years old in APOE ε4-related sporadic Alzheimer’s disease, and 85 years old in APOE *ε*4-unrelated sporadic Alzheimer’s disease. Compared with autosomal dominant Alzheimer’s disease, APOE *ε*4-related sporadic Alzheimer’s disease and APOE *ε*4-unrelated sporadic Alzheimer’s disease exhibit less aggressive Aβ and tau pathological changes over time. However, the three variants of Alzheimer’s disease consistently present pathological processes that may occur several decades before dementia onset [34,39,40,41].

Very early Aβ/tau pathologies in brain tissues have been confirmed by accumulating evidence. In carriers of autosomal dominant mutations linked to Alzheimer’s disease, cerebrospinal fluid (CSF) Aβ42 decline can occur 25 years before expected symptom onset, positron emission tomography (PET)-detected Aβ deposition is found 15 years before expected symptom onset, and CSF tau elevation can be found 15 years before expected symptom onset [11]. In non-demented young individuals, pathologically phosphorylated tau was detected in subcortical nuclei at 6 years old, and one 17-year-old subject exhibited extracellular Aβ plaque and intraneuronal phosphorylated tau in brain tissues [42]. An increased CSF phosphorylated tau (p-tau) 181 concentration, a decreased CSF Aβ 42/40 ratio, bilateral hippocampus atrophy and bilateral temporal lobe hypometabolism detected via PET-MRI (magnetic resonance imaging) were found in a 19-year-old patient diagnosed with probable Alzheimer’s disease [43].

Cognitive deficits can be found in such an early stage of Aβ/tau pathologies. For people with a first-degree family history of Alzheimer’s disease, verbal learning and memory function may be slightly impaired as young as at 18 years old, approximately 40 decades before the onset of sporadic Alzheimer’s disease [44]. Visuospatial and constructional skills may be impaired as early as at the age of 11–16 years if an individual has both positive APOE *ε*4 and a family history of Alzheimer’s disease [45]. For carriers of autosomal dominant Alzheimer’s disease, cognitive decline can occur around two decades before expected symptom onset (Figure 2).

## 3. Strategies for Preventing Dementia Due to Alzheimer’s Disease

Downward trends in the incidence/prevalence of dementia have occurred in developed countries although effective therapies for blocking pathological processes of Alzheimer’s disease are still lacking. The prevalence of dementia among the elderly over 65 years old in the US decreased from 11.6% to 8.8% from 2000 to 2012, and a higher level of education was found to be associated with a reduced risk of dementia [47]. Short-term specific cognitive training provided to more than 2000 elderly people (average age: 74 years) was found to evoke cognitive protection effects for 10 years [48]. A study investigating around 200,000 elderly people over 60 years old in the UK found that genetic factors and lifestyle contributed to dementia onset, whereas even for those with highly pathogenic genes, a healthy lifestyle (not smoking, exercise and a healthy diet) reduced dementia risks [49]. Exercise can ameliorate Alzheimer’s disease by regulating the immune response of the central nervous system and promoting hippocampal neurogenesis [50]. Adults performing moderate to vigorous physical activities no less than once each week exhibited a 34–50% risk reduction in cognitive decline and dementia with a duration of 8 to 10 years [51]. In mutation carriers of autosomal dominant Alzheimer’s disease, high-level physical activities (at least 150 min per week) are found to improve cognitive and functional performance, reduce the level of cerebral tau and Aβ pathology, and delay symptom onset by 15 years compared with low-level physical activities (less than 150 min per week) [52]. Even in older adults with mild-to-moderate Alzheimer’s disease dementia, moderate-intensity cycling and light-intensity stretching can reduce the progression of cognition decline [53]. On the contrary, middle-aged and elderly people living in the most vulnerable communities experiencing poverty and fewer educational and employment opportunities suffered accelerated brain atrophy and cognitive decline [54]. Additionally, the prevalence of dementia in rural areas with fewer medical and educational resources was approximately twice as high as that in urban areas [55].

These findings support that education, cognitive training, physical exercise/activities, and a healthy lifestyle can protect cognitive function. Additionally, educational, social and medical support should be provided to people living in poorer conditions. Recently, internet-delivered programs are expected to increase the accessibility of these interventions to prevent or delay cognitive decline [56]. In a cohort recruiting 2972 participants with a mean age of 52 years, an internet-based, self-motivated lifestyle intervention resulted in lifestyle changes in behavioral risk factors associated with cognitive decline (physical activity, diet, smoking, alcohol, sleep, and stress); the improvements lasted for over one year [57]. Another study also found that older adults with subjective cognitive decline receiving internet-delivered interventions for 52 weeks acquired statistically significant improvements in cognitive function, depression, and anxiety levels [58].

The protective effects contribute to human brain “resilience” [59] that may prevent, delay, or alleviate cognitive impairment due to Alzheimer’s disease or other neurological causes. Such resilience may be associated with high levels of education, socioeconomic status, and cognition-demanding daily tasks [60,61,62,63,64]. A study found that APOE *ε*4 carriers exhibited positive associations between years of education and brain metabolism in frontal and temporal cortices, and brain metabolism in these areas was related to better episodic memory function [65]. These findings indicate that higher levels of education help counteract deleterious effects of pathology on cognitive function such as episodic memory [66]. Elderly people with early-life high education show better multi-domain cognitive functioning, higher frequencies of participation in knowledge-related leisure activities, slower age-related reductions in executive function, and a larger gray matter volumes of the anterior brain regions [67]. Thus, childhood education should be prioritized because less education in early life has the highest prevalence among the 12 dementia risk factors except for air pollution (Figure 3) [68].

Thus, the first step is to promote awareness of dementia among the general public and medical professionals in non-developed countries as 50% of people with dementia in developed countries are diagnosed, compared with less than 10% in low- and middle-income countries [69].

Secondly, a healthy lifestyle and higher levels of education and exercise are recommended for reducing the risk of dementia. These measures do not require high-tech and costly drug development processes. For instance, elderly MCI patients with lower serum folate levels (mean age: 72.8 years) tended to progress to dementia [70], whereas lower serum folate levels can be corrected via dietary interventions [71,72]. On the contrary, drugs such as Lecanemab and Aducanumab are expensive (marketed at US $26,500 or 28,000 for a year’s worth of treatment) and inconvenient for patients due to infusion every few weeks being the route of administration [73].

Thirdly, Alzheimer’s disease is often associated with many comorbidities such as type 2 diabetes mellitus [4], cerebral and heart vascular pathologies [5], hypertension [6] and systemic infection [7]. Therefore, these chronic diseases should be adequately managed.

Fourthly, interventions targeting social isolation and disengagement should be adopted for dementia prevention. Weak social networks in late life (71–93 years) was associated a higher incidence of all-cause dementia and Alzheimer’s disease but not vascular dementia [74]. This finding underscores the need to promote social support for the elderly. In 865 community-based individuals aged 65 and above, social support and cognitive performance were positively related, and high levels of social support, particularly in support utilization, were related to low risks of cognitive impairment [75]. A systematic review comprising 2,370,452 participants from 33 studies indicates that poor social engagement (social network and social support) was associated with increased dementia risk whereas good social engagement was modestly protective when the follow-up duration was ≥10 years [76]. The Wisconsin Registry for Alzheimer’s Prevention study (1052 individuals with a mean age of 60.2 years) found that social support and verbal interaction were positively associated with different cognitive domains (social support: speed and flexibility; verbal interaction: verbal learning and memory). The author proposed that social support might counteract stress and that verbal interaction could enrich the environment and thus could be uniquely beneficial [77]. Generally, a lower total cerebral volume is associated with poorer cognitive function, which indicates an increased vulnerability to Alzheimer’s disease and related disorders at the preclinical stage; however, a large, population-based, longitudinal cohort study (2171 adults of a mean age of 63 years) indicates that social support, in the form of supportive listening, is associated with greater cognitive resilience and could modify such an association [78]. In this study, greater cognitive resilience is defined as better cognitive performance than estimated by lower total cerebral volume measured by magnetic resonance imaging [78], indicating that structural neurodegeneration is not necessarily parallel to cognition levels.

Some individuals exhibited a high burden of Aβ plaques and tau tangles upon autopsy and therefore would be expected to have had severe dementia, but these subjects remained at their cognitive baseline, indicating brain ‘resilience’ [79]. Such a striking mismatch between lesions and cognitive function provides hope for disease-modifying treatments for Alzheimer’s disease. When examining the multimodal association cortex lining the superior temporal sulcus, four main phenotypic features of individuals without experiencing dementia compared with demented Alzheimer’s cases with equivalent loads of Aβ plaques and tangles were revealed: (i) striking preservation of neuron numbers, synaptic markers and axonal geometry; (ii) significantly lower burdens of fibrillar thioflavin-S-positive plaques and of oligomeric Aβ deposits reactive to conformer-specific antibody NAB61; (iii) no strong and selective accumulation of hyperphosphorylated soluble tau multimers into the synaptic compartment (such accumulation was found in demented cases); and (iv) remarkably reduced glial activation accompanying Aβ and tau pathologies (the glial activation were robust in demented cases) [80]. Non-demented individuals with Alzheimer’s pathology were generally higher educated, had enhanced neural hypertrophy that could compensate for hippocampal and cingulate atrophy, showed decreased glial activation and reduced neuroinflammation, and exhibited increased levels of neural stem cells and ‘von Economo neurons’ (specialized nerve cells in a thicker region of the anterior cortex) [81]. These findings demonstrate the impact of factors other than the load of Aβ plaques and tau tangles on cognition performance in people with Alzheimer’s pathology.

## 4. Performing Cognitive Screening before MCI

Dementia risk factors should be identified and modified as early as possible. For instance, when examining risk factors for the pathology of Alzheimer’s and clinical–pathologic correlations, idea density and grammatical complexity of autobiographical essays written at a mean age of 22 showed a strong association with clinical outcomes in late life and the severity of Alzheimer’s neuropathology; individuals with idea density scores in the lowest tertile had more than 30 times the odds of having poor MMSE performance in late life compared with those in the upper two tertiles, and grammatical complexity also showed a very strong association (more than 16 times the odds). Notably, although lower attained education was not associated with the severity of Alzheimer’s neuropathology, it greatly increased the risk of dementia [82]. These findings denote the importance of education, which should be received in early life.

When Alzheimer’s pathology advances to the stage of mild cognitive impairment (MCI) or mild dementia, interventions may not be successful at blocking the development of the pathological process [83]. Compared with subjects with normal cognitive performance, MCI individuals reverting to normal cognitive function exhibited a high probability of progressing to dementia [84]. Therefore, it is necessary to take measures before the MCI stage [83].

Fluid and neuroimaging biomarkers used to detect early stages of Alzheimer’s pathology have been developed [68,85,86]. PET imaging is very costly. However, the prevalence of PET-detected Aβ positivity is only 2.7% in people aged 50–59 years without cognitive impairment, and the probability of developing Alzheimer’s disease in MCI cases with Aβ PET positivity (hazard ratio: 1.9) is similar to that of those who are Aβ-negative (hazard ratio 1.6) [87]. Low-cost blood biomarkers have wider acceptability and applicability. Plasma phospho-tau (p-tau) 217 is found to exhibit high performance when detecting abnormal Aβ status or progression to Alzheimer’s dementia in the elderly with MCI, and a high correlation level (R = 0.89) between plasma and CSF values was observed [88]. However, effectively identifying early, subtle cognitive dysfunctions prior to MCI in large populations still needs cognitive tests.

The 2018 NIA-AA research framework introduced six clinical stages in the Alzheimer’s continuum (Figure 4). As a stage prior to MCI, transitional cognitive decline features subjective reporting of cognitive decline, subtle decline upon longitudinal cognitive testing, or both a subjective report of decline and objective evidence of it upon longitudinal testing [3]. However, a quickly implemented screening tool for objectively measuring cognitive performance at this stage appears to be lacking, although assessing subjective cognitive decline is feasible.

Performance-based cognitive tests such as MMSE and the Montreal Cognitive Assessment (MoCA) are widely adopted in MCI screening [89], though MMSE is not as sensitive as MoCA for assessing MCI among elderly people [90]. An informant-reported instrument, AD8, was developed to differentiate non-demented subjects and very mild dementia [91], but its effectiveness for MCI screening may not be as good as that for dementia screening [92]. Another type of cognitive test, questionnaires to assess self-reported concerns on cognition such as the SCD-9 questionnaire (subjective cognitive decline, SCD) can be used to evaluate cognitive decline prior to MCI. However, SCD scores are related to demographic factors, e.g., females are more concerned than males, subjects with lower education levels are more likely to worry about their cognitive performance, and younger individuals might have more concerns than more elderly adults do [93]. CDR, a mixed measure containing three types of testing items (i.e., performance-based, informant-reported and subjective concerns), is widely used to assess cognitive and functional performance (0 indicates “no dementia”, 3 means “severe dementia”, and a global rating of 0.5 indicates MCI). The long duration of administration (possibly up to 90 min) makes CDR impracticable as a screening tool [94,95,96].

Behavioral biomarkers may be used in cognitive screening. When testing average stroke sizes, handwriting, a type of fine motor control, was found to progressively decline with human aging [97]. During non-alphabetical and alphabetical writing, Alzheimer’s disease patients showed less automated movements, decreased writing velocity and decreased frequency of up-and-down strokes [98]. Tremendous writing pressure and poorer writing stability were also found in patients with Alzheimer’s disease [99]. Compared with regular writing and multiply repeated single-letter writing, an all-capital-letters writing task was better for distinguishing MCI patients from healthy controls [100]. Digital handwriting analysis also can detect differences in handwriting features between MCI cases and age-matched healthy controls via the use of a task of writing a Chinese character [101]. Based on handwriting kinetics and quantitative electroencephalography (EEG) analysis, the differentiating result between MCI and healthy elderly controls could reach a classification result of 96.3% [102]. Other digital cognitive biomarkers, such as those measured with drawing tests, daily living tasks, and electronic games, may achieve comparable or even better diagnostic performance than classical paper-and-pencil tests for detecting MCI and dementia [103] but the ability of these methods to detect transitional cognitive decline is still unknown.

Cognitive tests (e.g., MMSE and AD8) designed to detect very mild dementia or MCI are not sensitive enough to identify cognitive decline prior to MCI [104]. Additionally, many cognitive tools do not adequately take into account the cross-language and cross-cultural consistency of testing results. For instance, the Boston Naming Test ranks tested items by frequency, from “bed” to “abacus” [105]. However, there may be a huge difference in frequencies of the same term in different areas/countries; it is difficult to identify an abacus in the UK and USA but the item is easy to be recognized in some Asian countries such as China. As another example, some proverbs (e.g., “A stitch in time saves nine”) are difficult to be translated across languages. Even the developer of MMSE is not certain as to the appropriate translation of “No ifs, ands, or buts” [106]. This means that some parts of a cognitive test in different language versions actually examine distinct contents, and the thus testing results of an English version may not be well-comparable with the results using non-English versions. When examining Hebrew-speaking elderly persons in three different ways in the MMSE repetition task “No ifs, ands, or buts” (a literal translation of the English sentence, a well-known Hebrew proverb consisting of monosyllabic words and rhythmic effects, and another well-known Hebrew proverb without such features), different predictive values did exist [107].

A cognitive test is usually designed with a “building blocks” approach. For example, MMSE contains three-word recall, time and place orientation, three-object naming, sentence repetition, following verbal and written commands, writing a sentence, subtraction (serial 7s) and copying intersecting pentagons [108]. These items come from the mentor of the author and textbooks [106]. MoCA uses five-word recall rather than the three-word memory test used in MMSE. Additionally, MoCA supplemented several additional components including the copy of cube test, trail making test, random letter test, and clock drawing test, which present higher challenge levels than does in MMSE. The copy of cube test requires the subject to draw an identical cube including parallel lines and right angles [109], whereas copying intersecting pentagons only requires presenting all 10 angles with 2 angles intersecting [108]. The supplemented tests in MoCA contribute to its increased sensitivity compared to MMSE [90,110].

For completing copying or drawing tests, individuals have to use their visuoconstructional and executive functions. Visuospatial function impairments cause difficulties in spatial orientation during movements, problems in spatial coordination, perception of the relative size of objects, and spatial distances and positions [111]. The trail-making test comprises two components: Part A to connect numbers with lines in a numerical sequence (i.e., 1–2–3–…) and Part B to draw lines connecting numbers and letters in an alternating numeric and alphabetic sequence (i.e., 1–A–2–B–3–C–…). Part A is a visuomotor sequence tracking task whereas Part B additionally examines the flexibility of set-switching between numbers and letters. Variance in time to completing Part B is uniquely impacted by visuomotor scanning, whereas working memory and executive functioning contribute to the error score of Part B [112]. The completion time of Part B can be used to differentiate healthy elderly people from MCI patients but with low sensitivity [113]. Similarly, a low sensitivity for MCI vs. healthy aging was also revealed in another study where the area under the receiver operating characteristic (ROC) curve was only 0.70 for Part B [114].

Orientation is a traditionally defined cognitive domain and is often included in screening cognitive tests such as MMSE and MoCA. MCI cases with MMSE-detected orientation deficits had around 1.5 times the risk of Alzheimer’s disease progression when compared with oriented subjects, and temporal disorientation (18%) but not spatial disorientation (9%) accounted for most of the explained variance [115]. Two temporal orientation items in MMSE, date and day of week, were found to be inversely associated with Alzheimer’s disease progression in MCI cases whereas other temporal orientation items (i.e., year, month, and season) and the five spatial orientation items could not provide information for prediction [116].

Spatial disorientation such as getting lost is considered one of the earliest signs of Alzheimer’s disease [117]. Grid cells, head-direction cells, border cells and other neurons in the medial entorhinal cortex participate in the construction of spatial cognitive maps in the brain [118,119,120,121]. The spatial cognitive map is a representation of the external environment in the human brain, which may involve several aspects: the entorhinal cortex and hippocampus are responsible for map-like spatial coding, posterior brain regions (e.g., parahippocampal and retrosplenial cortices) support the anchoring of the cognitive maps to environmental landmarks, and the spatial map encoded by hippocampal and entorhinal spatial codes works with the frontal lobe to complete route planning during navigation [122]. In Alzheimer’s disease patients, tau protein is deposited in the medial temporal lobe [23,24], where the atrophy of the hippocampus and parahippocampal gyrus occurs and affects spatial navigation function [123]. As an important cognitive function, spatial navigation involves searching, planning, identifying and maintaining paths. Spatial navigation dysfunction leads to topographical disorientation and getting lost [124]. Spatial navigation declines with age [125,126] and performances of spatial navigation measured by the Floor Maze Test worsened with increasing severity of cognitive impairment from subjective cognitive impairment to MCI and mild Alzheimer’s disease [124].

Taken together, new tools are needed for performing cognitive screening to detect transitional cognitive decline, a stage prior to MCI. Many cognitive domains may be examined to detect mild cognitive deficits in this stage.

## 5. Conclusions

Alzheimer’s pathology develops over a long duration lasting several decades. Compared with MCI, an earlier stage, transitional cognitive decline, may be a better time window in which effective interventions can be adopted. Detecting this stage in large populations relies on the rapid screening of cognitive function. Given that many cognitive tests focus on MCI detection, new tools need to be developed.

## Figures and Tables

**Figure 1 biomedicines-11-01423-f001:**
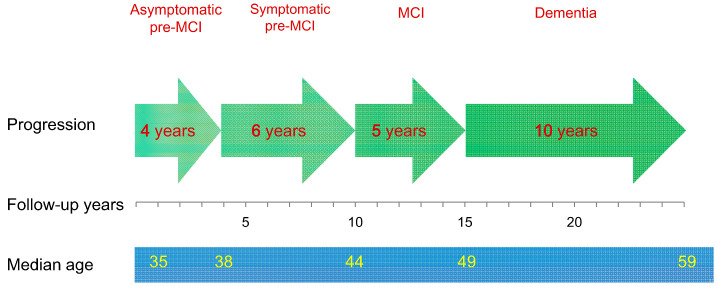
Progression courses of Alzheimer’s disease in PSEN1 carriers. MCI: mild cognitive impairment. Median onset age: 35 years for asymptomatic pre-MCI, 38 years for symptomatic pre-MCI, 44 years for MCI, and 49 years for dementia. Median time of progression from asymptomatic pre-MCI to dementia: 15 years (from asymptomatic to symptomatic pre-MCI: 4 years; from symptomatic pre-MCI to MCI: 6 years; from MCI to dementia: 5 years; and from dementia to death: 10 years) [19]. A late stage is indicated in a darker color.

**Figure 2 biomedicines-11-01423-f002:**
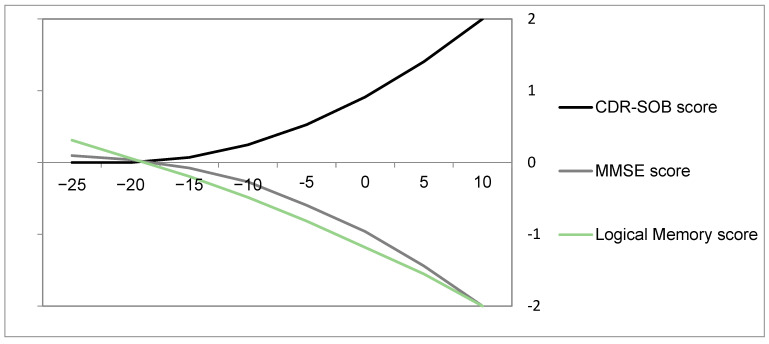
Cognitive decline occurs around two decades before expected symptom onset in autosomal dominant Alzheimer’s disease. The *y*-axis indicates the normalized differences between mutation carriers and non-carriers (the value of carriers minus the value of non-carriers); during normalization, the variation across all eight time points is converted to be in the range of [–2, 2] for each cognition parameter, and 0 in the *y*-axis means the same level between carriers and non-carriers (difference = 0). The *x*-axis indicates estimated years from expected symptom onset; 0 in the *x*-axis indicates the year of symptom onset and −20 means 20 years prior to symptom onset. The estimated years from expected symptom onset are acquired by using the age of the participant receiving study assessments to minus the age of the parent at dementia onset, by which the disease trajectory over time can be approximated [11,46]. The curves are plotted with the point-fold line chart using the data from [11]. Mini-Mental State Examination (MMSE), Logical Memory subtest of the Wechsler Memory Scale–Revised, and Clinical Dementia Rating–Sum of Boxes (CDR–SOB) measure cognitive function, and higher MMSE/Logical Memory test scores indicate better function whereas higher CDR–SOB scores indicate worse function; thus, the downtrending MMSE/Logical Memory and uptrending CDR–SOB curves consistently represent the greater cognitive decline caused by mutation carriers. All curves consistently show a decline in cognition starting around two decades before symptom onset.

**Figure 3 biomedicines-11-01423-f003:**
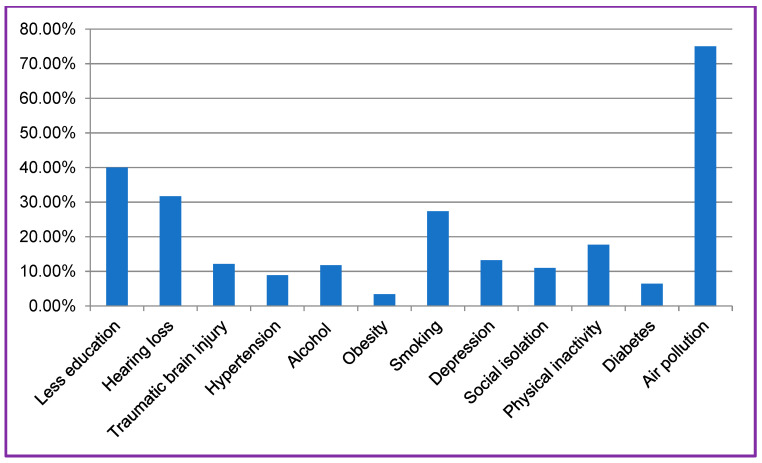
Prevalence of risk factors for dementia. Among the 12 potentially modifiable risk factors for dementia, less education in early life (<45 years) has the highest prevalence (except air pollution) [68].

**Figure 4 biomedicines-11-01423-f004:**
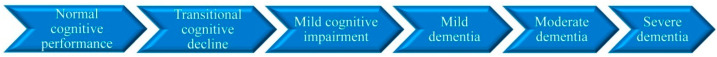
Stages in the Alzheimer’s continuum. Alzheimer’s pathologic change (abnormal Aβ biomarkers with normal pathologic tau biomarkers) and Alzheimer’s disease (abnormal Aβ and pathologic tau biomarkers) are not separate entities but earlier and later phases/stages of a continuum. Stage 1: normal cognitive performance; Stage 2: transitional cognitive decline; Stage 3: MCI, Stage 4: mild dementia; Stage 4: mild dementia; Stage 5: moderate dementia; and Stage 6: severe dementia [3].

## Data Availability

Not applicable.

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
