# Peer review of "Ultra-Early Screening of Cognitive Decline Due to Alzheimer’s Pathology"

_biomedicines, 2023, doi:10.3390/biomedicines11051423_

Round 1

Reviewer 1 Report

Dear Editor,

I carefully read the manuscript ""Transitional cognitive dysfunction” or “Transitional cognitive decline” for ultra-early screening of Alzheimer’s disease? by Prof. Wei.

My comments and suggestions for the Author are the following:

 - All the abbreviations should be defined at their first appearance in the manuscript.

 - Tables: All the abbreviations used in the tables should be defined at the bottom of the tables.

 - The author should more carefully specify in the manuscript how the selection process of literature was carried out.

 - References should be reported in the manuscript following the Instructions for Authors of the Journal.

 - Figures: Authors should give titles to graph axes.

 - Lines 468-472: These sentences should be moved to the Conclusion paragraph.

- Line 493: A reference is missing here.

- Line 497: A reference is missing here.

- Line 501: A reference is missing here. 

- Line 511: A reference is missing here.

 - Line 514: A reference is missing here

 - Some references are obsolete and should be replaced by more recent articles.

Please, see my comments below.

Author Response

Many thanks for the suggestions.

Abbreviations have been defined at their first appearance in the manuscript.

Selection of the literature is based on the contents. For instance, when introducing Aβ and tau biomarkers, relevant literature are selected.

The graph axes of Figures have been introduced in the figure legends.

Some contents have been modified, and new references have been added.

Reviewer 2 Report

Thank ou for permitting me to review this manuscript

The title appears to focus on early stage of AD , some sentence explaining the objective of this review/commentary , should be provided at the end of the introduction 

Introduction : Line 47-49 , please delete  there is no point to this , there is not enough data to justify this alert 

Line 82 may be I missed something , what is iii? 

Line 174-178 please provide reference (PPR)

Pleas elaborate somwhere in the text brain hyperactivity and better cognitive performance 

Figure 1 : please explain clearly which data was used in this graphic (data source) 

Line 473-77 PPR

Line 515 , a change in a vocabulary should better come  at least  after an international experts meeting , otherwise it might promote confusion , this should not be stated in the conclusion , may be it is better to state this previous chapters.

Author Response

Many thanks for the suggestions.

The introduction section has been modified.

Contents in Line 47-49, Line 82, and Line 174-178 have been adjusted.

Contents on brain hyperactivity and better cognitive performance have been replaced by cognition testing methods.

The source of each Figure has been noted by the cited reference. In the legend of each figure, one or more reference is provided. The data comes from the reference.

Reviewer 3 Report

Review of a manuscript “Transitional cognitive dysfunction” or “Transitional cognitive decline” for ultra-early screening of Alzheimer’s disease? By Pengxu Wei submitted to “Biomedicines”

Alzheimer’s disease is the most frequent neurodegenerative disease imposing enormous damage to the patients, their relatives and to the healthcare system all over the world. Unfortunately, there is no efficient disease-modifying therapy that changes the main course of this disorder. The author reviews and discusses the major types of defining and diagnosing Alzheimer’s disease. This is an important biomedical issue and the results presented in the manuscript will be interesting for the readership of “Biomedicines”.

The following corrections and additions should be made.

Abstract:

Line 12.  “Alzheimer’s disease imposes a severe burden.” This sentence is not in appropriate place and we suggest to delete it.  

Introduction

Line 37: “Dementia is a very costly disease because there has been no effective therapy and the patients need expensive care for years [1, 2].” After this sentence we recommend to add the following sentence and a reference: ”AD is often associated with comorbidities which aggravate the course of the disease {reference “Caveolin: A New Link Between Diabetes and AD. Cell Mol Neurobiol. 2020 Oct;40(7):1059-1066. doi: 10.1007/s10571-020-00796-4.”

Lines 55, 57  “neurodegeneration [AT (N)]. “ The author should give a brief explanation of AT (N)].

Lunes 89-115 For a manuscript with a title “Transitional cognitive dysfunction” or “Transitional cognitive decline” a detailed description of different molecular forms of Aβ polypeptide looks too long. It should be more concise  and succinct.

Lines 116-127 – the same concerns detailed description of apolipoprotein E (APOE), which are also well known and should be presented in a more succinct way.

Lines 144-145. “after the early tau pathology, Aβ deposition occurs independently and then induces acceleration of tauopathy.” Reference should be added for this statement.

Lines 183-186. Figure 1. “(I) The y-axis indicates the normalized differences between mutation carriers and non-carriers…” It is unclear how the Roman figures I, II, II presented in the Figure legend can be found on the figure. Clarifications needed.

Lines 473-474:” Such hyperactivation in brain regions in young individuals at risk of Alzheimer’s disease may be caused by toxicological stimuli, e.g., Aβ-glutamate interaction or fatty acids.” References on this statement should be added.

7. Conclusion

Line 505. “The preclinical period of Alzheimer’s disease may be a long period.” The sentence should be corrected as “The preclinical period of Alzheimer’s disease may be long.”

Overall, an interesting manuscript containing new hypothesis and stimulating discussion.

Author Response

Many thanks for the comments and suggestions.

The content of Line 12 has been adjusted. 

Line 37: The reference has been supplemented.

Lines 55, 57  “neurodegeneration [AT (N)]. These words have been removed in the new version.

Lunes 89-115 The title of the manuscript has been adjusted, and the discussion of molecular forms of Aβis short in the new version.

Lines 116-127 –APOE is also introduced concisely in the new version.

Lines 144-145. Reference supporting earlier tau pathology or earlier Aβ deposition has been added.

Lines 183-186. This figure has been modified to be simple. I, II, II have been removed.

Lines 473-474: Contents on hyperactivation have been removed.

To sum up, many modifications have been made to avoid ambiguity in the new version.